# Predictive models for secondary epilepsy in patients with acute ischemic stroke within one year

Jinxin Liu[1†], Haoyue He[1,2†], Yanglingxi Wang[1], Jun Du[3], Kaixin Liang[4], Jun Xue[5], Yidan Liang[1], Peng Chen[1], Shanshan Tian[6*], Yongbing Deng[1,7,8*]

[1]Department of Neurosurgery, Chongqing Emergency Medical Center, Chongqing University Central Hospital, School of Medicine, Chongqing University, Chongqing, China; [2]Bioengineering College of Chongqing University, Chongqing, China; [3]Department of Neurosurgery, Chongqing University Qianjiang Hospital, Chongqing, China; [4]Department of Neurosurgery, Yubei District Hospital of Traditional Chinese Medicine, Chongqing, China; [5]Department of Neurosurgery, Bishan hospital of Chongqing Medical University, Chongqing, China; [6]Department of Prehospital Emergency, Chongqing University Central Hospital, Chongqing Emergency Medical Center, Chongqing, China; [7]Chongqing Key Laboratory of Emergency Medicine, Chongqing, China; [8]Jinfeng Laboratory, Chongqing, China

*For correspondence:
710836163@qq.com (ST);
dyb0913@cqu.edu.cn (YD)

[†]These authors contributed equally to this work

Competing interest: The authors declare that no competing interests exist.

## eLife Assessment

This **valuable** study reports machine learning models derived from large-scale data to predict the risk of post-stroke epilepsy. The evidence supporting the conclusions is **convincing**, although there are some validation issues (lack of cross-validation, possible bias in external validation results). The study may be of interest in the field of clinical neurology

## Abstract

**Background:** Post-stroke epilepsy (PSE) is a critical complication that worsens both prognosis and quality of life in patients with ischemic stroke. An interpretable machine learning model was developed to predict PSE using medical records from four hospitals in Chongqing.

**Methods:** Medical records, imaging reports, and laboratory test results from 21,459 ischemic stroke patients were collected and analyzed. Univariable and multivariable statistical analyses identified key predictive factors. The dataset was split into a 70% training set and a 30% testing set. To address the class imbalance, the Synthetic Minority Oversampling Technique combined with Edited Nearest Neighbors was employed. Nine widely used machine learning algorithms were evaluated using relevant prediction metrics, with SHAP (SHapley Additive exPlanations) used to interpret the model and assess the contributions of different features.

**Results:** Regression analyses revealed that complications such as hydrocephalus, cerebral hernia, and deep vein thrombosis, as well as specific brain regions (frontal, parietal, and temporal lobes), significantly contributed to PSE. Factors such as age, gender, NIH Stroke Scale (NIHSS) scores, and laboratory results like WBC count and D-dimer levels were associated with increased PSE risk. Tree-based methods like Random Forest, XGBoost, and LightGBM showed strong predictive performance, achieving an AUC of 0.99.

**Conclusions:** The model accurately predicts PSE risk, with tree-based models demonstrating superior performance. NIHSS score, WBC count, and D-dimer were identified as the most crucial predictors.

**Funding:** The research is funded by Central University basic research young teachers and students research ability promotion sub-projec t(2023CDJYGRH-ZD06), and by Emergency Medicine Chongqing Key Laboratory Talent Innovation and development joint fund project (2024RCCX10).

## Introduction

Stroke is the second leading cause of death worldwide, with an annual mortality of about 5.5 million, and is the leading cause of disability, accounting for 50% of cases globally (*Feigin et al., 2017*). Ischemic strokes make up approximately 80% of all stroke cases (*Galovic et al., 2018*; *Krishnamurthi et al., 2013*). PSE is a common complication, with studies showing that 3–30% of stroke patients develop epilepsy, which worsens their prognosis and quality of life (*Zhao et al., 2018*). PSE can exacerbate cognitive, psychiatric, and physical impairments caused by cerebrovascular disease and related conditions (*Al-Sahli et al., 2023*). The highest incidence of PSE occurs within the first year after an acute stroke, representing nearly half of all cases (*Galovic et al., 2018*). Therefore, early prediction and intervention for PSE, especially in ischemic strokes, are essential.

Currently, most studies rely on clinical data to build statistical models using survival analysis, Cox regression (*Galovic et al., 2018*; *Chen et al., 2018*), and multiple linear regression (*Merkler et al., 2018*) to predict PSE. Last year, *Lin et al., 2023* developed a radiomics-based model that outperformed conventional clinical models in predicting PSE related to intracerebral hemorrhage (ICH), suggesting that a combined radiomics-clinical model could improve individual risk assessment of PSE after the first ICH, facilitating early diagnosis and treatment. However, later research raised concerns about the use of radiomics, indicating the need for further investigation (*Pszczolkowski and Law, 2023*). Overall, research on PSE prediction remains limited, with most studies focusing on specific risk factors (*Waafi et al., 2023*; *Herzig-Nichtweiß et al., 2023*; *Lin et al., 2023*; *Pitkänen et al., 2016*) and building basic models, rather than proposing more comprehensive and scientifically robust prediction models.

Machine learning has gained recognition as a powerful tool for developing medical models, due to its ability to process large datasets and handle complex information. It has been increasingly applied in neuroscience and clinical prediction (*El Naamani et al., 2024*; *Lu et al., 2023*; *Daidone et al., 2024*). Previous studies have used machine learning to explore post-stroke cognitive impairments (*Lee et al., 2023*), predict stroke and myocardial infarction risk in patients with large artery vasculitis (*Lu et al., 2023*), develop post-stroke depression models using liver function tests (*Gong et al., 2023*), and predict hematoma expansion in traumatic brain injury (TBI) (*He et al., 2024*). Machine learning models can automatically capture both linear and complex nonlinear relationships between variables, providing insights into how different factors contribute to the prediction target—something traditional statistical models struggle with. However, machine learning requires large datasets and is prone to overfitting when sample sizes are small. The quality and quantity of input data are critical for the algorithm to detect patterns and make accurate predictions.

This study aims to identify key risk factors from features extracted from clinical records and test data of ischemic stroke patients. Using these features, a machine learning-based model will be developed to predict PSE. By leveraging early admission data, the goal is to automatically predict the likelihood of PSE occurrence, guiding clinical decision-making and patient care.

## Results

### Filling of missing data

Missing values were imputed using a Random Forest (RF) model, addressing one feature at a time. The imputed features included: Plt, WBC, RBC, HbA1c, CRP, TG, LDL, HDL, AST, ALT, bilirubin, albumin, urea, creatinine, BUA, PT, APTT, TT, INR, D-dimer, fibrinogen, CK, CK-MB, LDH, HBDH, IMA, lactate, anion gap, TCO2, and NIHSS (*Figure 1*).

### Characteristics of study participants

A total of 21,459 patients were included in the study. The training set comprised 15,021 patients, with a PSE incidence of 4.3%. The test set consisted of 6438 patients, also with a 4.3% incidence of PSE. An

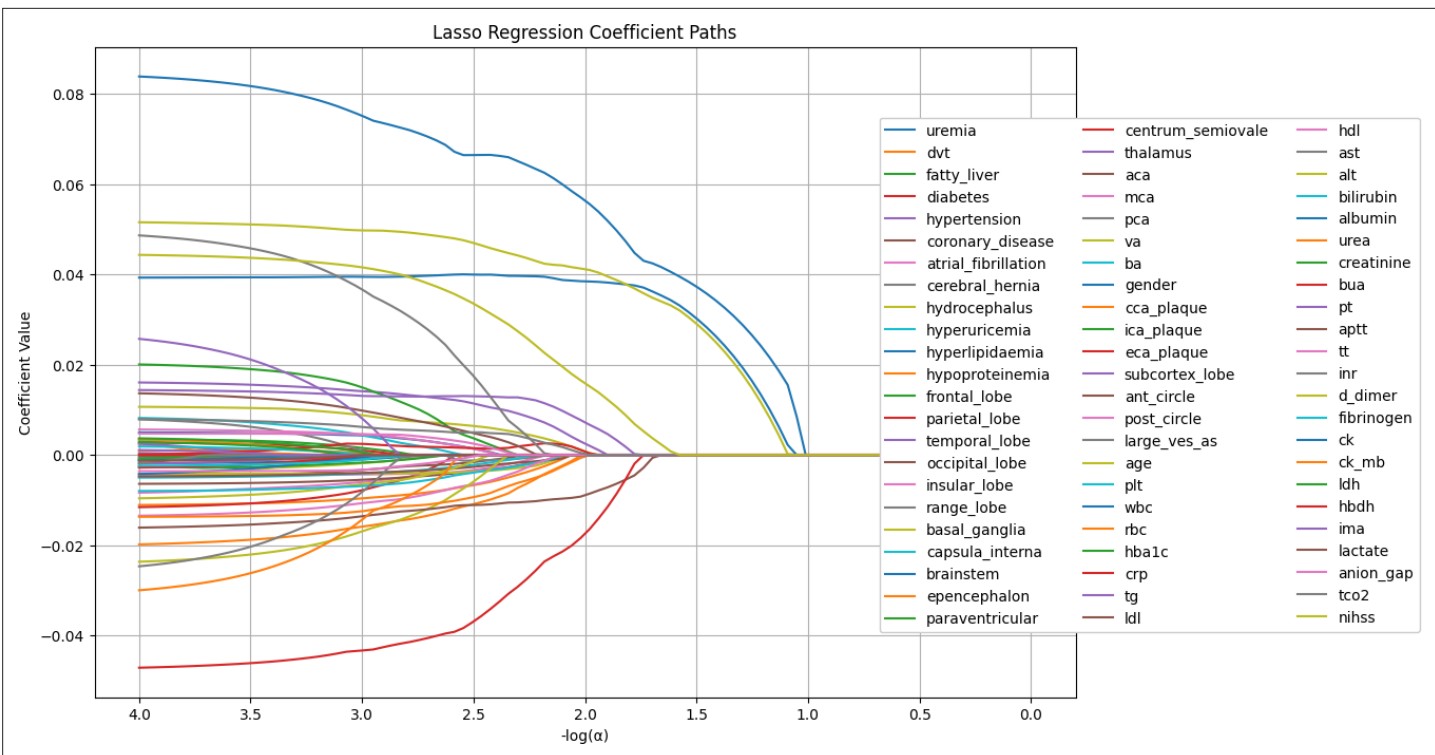

**Figure 1.** LASSO regression coefficient paths. The image shows the LASSO regression coefficient paths for various features related to a medical or research study. The x-axis shows the log of the regularization parameter alpha, and the y-axis shows the regression coefficient values. The lines in the plot represent the coefficient paths for different features as the regularization parameter changes. The features are labeled on the right side of the plot, and the most important features selected by the LASSO model are shown at the bottom of the image.

external validation cohort included 536 patients from three hospitals. Detailed statistical information on the clinical characteristics is presented in (*Supplementary file 1*).

Analysis indicated that patients with a higher likelihood of developing PSE had complications such as uremia, a history of DVT, atrial fibrillation, hyperuricemia, cerebral hernia, and hydrocephalus. Affected brain regions included the frontal, parietal, occipital, and temporal lobes, along with the cortex, subcortex, basal ganglia, and hypothalamus. General characteristics influencing risk included age, gender, and NIHSS. Laboratory indicators associated with a higher risk of PSE included WBC, HbA1c, CRP, TG, AST, ALT, bilirubin, urea, uric acid, APTT, PT, D-dimer, CK, CK-MB, LDH, HBDH, IMA, lactate, and anion gap. Significant p-values were also observed for fatty liver, coronary heart disease, hyperlipidemia, and HDL, with low or negative values linked to higher risks of secondary complications. Details of statistical, univariate, and multivariate regression analyses are provided in *Supplementary file 1*, *Supplementary file 2*, *Supplementary file 3*.

## Performance of machine learning models

Performance indicators for the machine learning models are summarized in *Supplementary file 4*. The ROC curves, calibration curve, and decision curve analysis (DCA) are shown in *Figure 2*. Tree-based models such as RF, XGBoost, and LightGBM demonstrated the highest AUC scores, outperforming other models. Among them, RF achieved the highest positive predictive value (PPV) at 0.864, which was the most significant metric in this study. Complex machine learning algorithms outperformed traditional logistic regression. The calibration curve showed a Brier score of 0.006, and DCA indicated strong practical value in clinical decision-making. In the external validation cohort, RF achieved a sensitivity of 0.91 and a PPV of 0.95, confirming its strong predictive capability.

## Analysis of SHAP risk factors

*Figure 3* presents SHAP values, individual decision attempts, and overall decision curves. Among general characteristics, females had a higher PSE rate. A higher NIHSS was associated with an

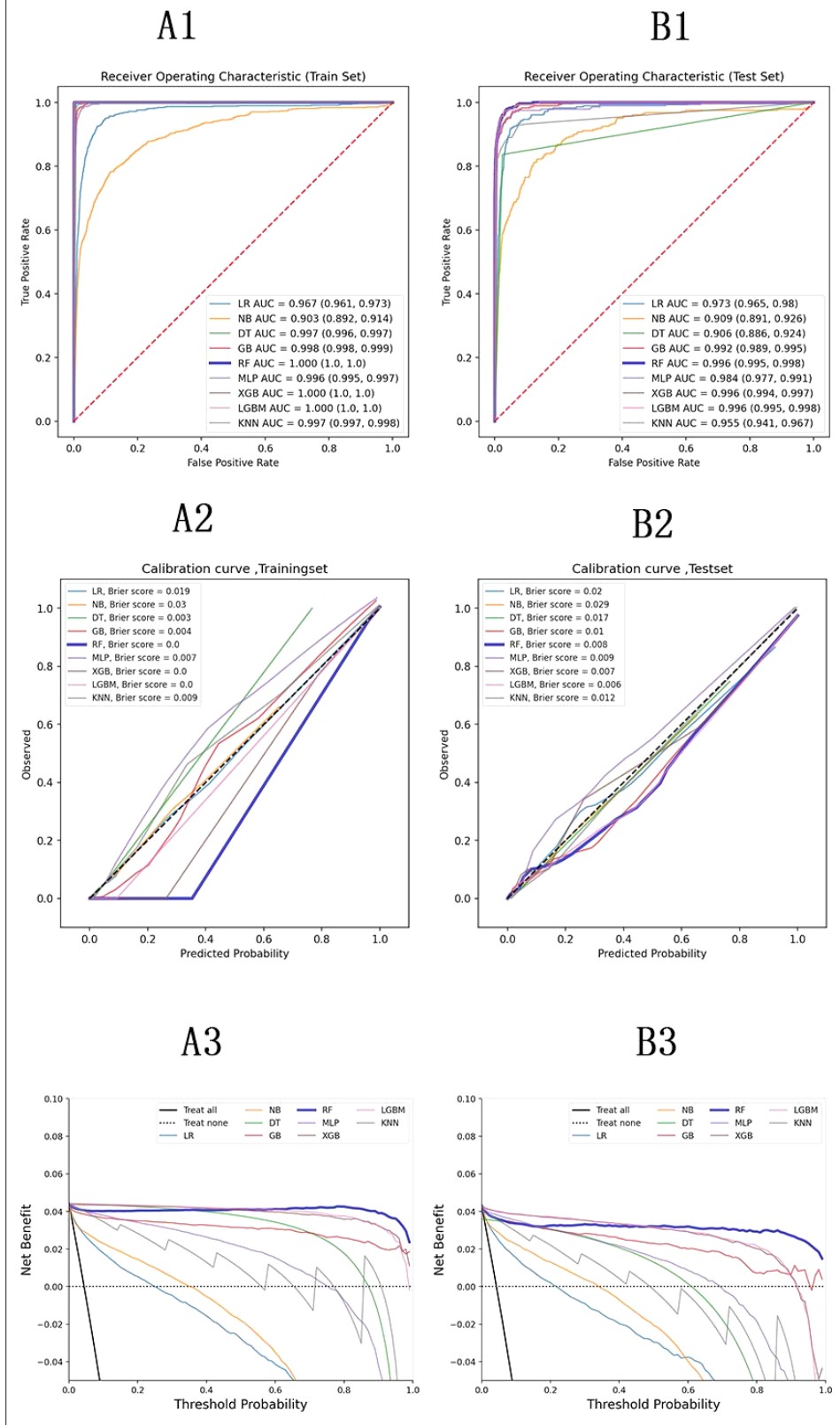

**Figure 2.** Model evaluation metrics and curves. The figure shows model performance curves across six sections (A1, A2, A3 on the left; B1, B2, B3 on the right) for training and test sets. ROC Curve: Illustrates the trade-off between sensitivity and specificity, with the AUC indicating overall model performance. Calibration Curve: Compares predicted probabilities to actual outcomes, assessing the model's confidence accuracy. Precision-Recall

*Figure 2 continued on next page*

*Figure 2 continued*

Curve: Analyzes the balance between precision and recall at various thresholds, particularly useful for imbalanced datasets.

The online version of this article includes the following figure supplement(s) for figure 2:

**Figure supplement 1.** ROC curve of train and test groups.

**Figure supplement 2.** Calibration curve of train and test groups.

**Figure supplement 3.** Precision-recall curve of train and test groups.

increased incidence of PSE. Elevated WBC, D-dimer, CRP, AST, CK-MB, HbA1c, bilirubin, TCO2, and LDH levels at admission were linked to a greater likelihood of developing PSE. Conversely, lower levels of HBDH, Plt, and APTT were also associated with a higher risk of PSE. Specific brain regions did not have a significant individual effect on the outcome. Among complications, hypertension was more strongly associated with PSE development, while conditions such as coronary heart disease, diabetes, hyperlipidemia, and fatty liver were less likely to be related. A force plot for the first patient illustrated the influence of different features on the prediction. In this case, a prolonged APTT time contributed the most to PSE, followed by elevated AST levels, while a low NIHSS negatively impacted the final result. The decision plot aggregated model decisions to demonstrate how complex models generated their predictions.

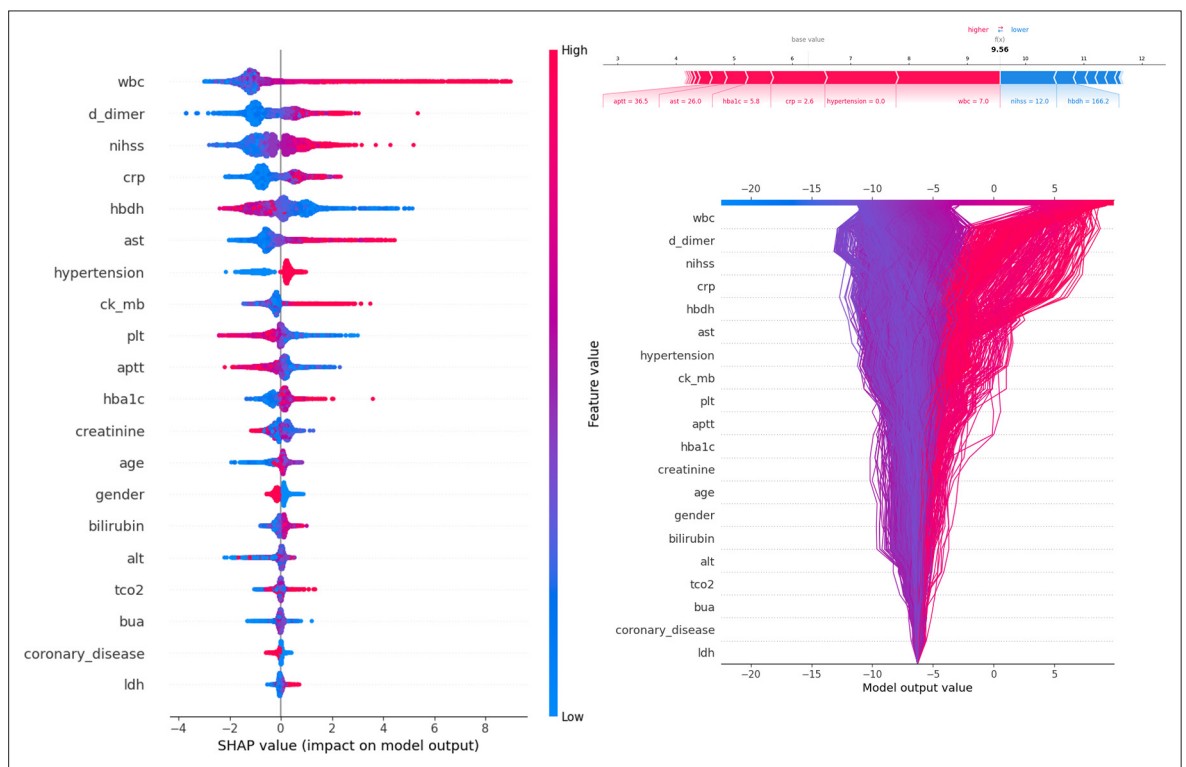

**Figure 3.** Description of the SHapley Additive exPlanations (SHAP) values and feature importance. SHAP Value (Left): Displays the impact of each feature on the model's predictions, with features sorted by importance. The color gradient indicates the range of feature values, from low (blue) to high (red). Force plot (upper right): Illustrates the contribution of individual features of the first sample to the final model output, highlighting how each feature value pushes the prediction away from the baseline value. Decision plot (lower right): Visualizes the cumulative impact of features on the model output for each sample, showing how the feature values combine to produce the final prediction.

The online version of this article includes the following figure supplement(s) for figure 3:

**Figure supplement 1.** SHAP value of train and test groups.

**Figure supplement 2.** Force plot of train and test groups.

**Figure supplement 3.** Decision plot of train and test groups.

## Discussion

The study utilized comprehensive clinical, imaging, and laboratory data from stroke patients to develop a predictive model using machine learning algorithms. The model achieved an AUC above 0.95, indicating more accurate predictions compared to traditional statistical methods. Tree-based ensemble models demonstrated superior predictive performance, particularly when handling large datasets with high-dimensional features.

During the modeling process, the extreme imbalance between negative and positive samples was addressed using the SMOTEENN technique, which improved model performance. SHAP analysis was conducted to assess model interpretability and identify the importance of different features.

According to the statistical results, age and NIHSS were treated as continuous variables. The results show that female patients, older individuals, and those with higher NIHSS were more likely to develop PSE, consistent with recent studies. Higher NIHSS, indicating more severe strokes, significantly increased the risk of complications, ranking second only to WBC and D-dimer in predictive importance (*Al-Sahli et al., 2023*; *Lin et al., 2021*; *Waafi et al., 2023*; *Zöllner et al., 2020*). However, there are differing perspectives regarding the effect of age. Some studies (*Al-Sahli et al., 2023*; *Yamada et al., 2020*) suggest that age below 65 is a high-risk factor, which aligns with this findings, while other studies (*Lidetu and Zewdu, 2023*) indicate that advanced age is the key factor. *Yamada et al., 2020* supported the results, showing that female patients have a higher risk of complications, while (*Waafi et al., 2023*) reported that male patients are 3.325 times more likely to develop complications, contradicting these results.

Previous research indicates that patients with diabetes, dyslipidemia, hypertension, depression, or dementia are at higher risk of developing vascular epilepsy (*Pitkänen et al., 2016*). In this study, statistical analysis and multiple ML models examined the relationship between comorbidities and complications. Patients with coronary heart disease, diabetes, fatty liver, hyperlipidemia, or large artery stenosis or plaques (CCA and ICA) were found to be less likely to develop epilepsy. According to the TOAST classification, ischemic stroke is divided into five categories: large artery atherosclerosis, cardioembolism, small vessel occlusion, other determined etiology, and undetermined etiology. Patients with multiple comorbidities often fall into the large artery atherosclerosis and cardioembolism categories, which are more clearly defined and easier to treat, resulting in a lower likelihood of epilepsy. In contrast, strokes of undetermined etiology tend to have worse prognoses and a higher likelihood of leading to epilepsy. Among patients with diabetes, higher HbA1c levels indicate poor blood sugar control and a higher risk of complications. Better blood sugar control was associated with a lower overall risk of developing complications.

*Lekoubou et al., 2023* reported that cortical infarction is more likely to lead to epilepsy in patients hospitalized with anterior circulation ischemic stroke. *Lin et al., 2023* also found that factors such as cortical involvement and intracerebral hemorrhage volume increase the likelihood of PSE, aligning with the statistical result findings. Al-Sahli et al. suggested that cortical brain injury and large-area lesions elevate the risk of PSE (*Al-Sahli et al., 2023*; *Yamada et al., 2020*). According to the statistical results, both cortical and subcortical involvement were associated with an increased likelihood of PSE, but these regions had a smaller influence compared to other features and were not selected in LASSO regression.

Previous studies have identified acute infection as a risk factor for ischemic stroke (*Bova et al., 1996*). CRP, which reflects inflammation levels, is an independent prognostic factor (*Di Napoli et al., 2001*). Both regression and SHAP analysis of the results indicated that WBC had a significant impact among routine blood test parameters, even surpassing NIHSS in SHAP analysis. A high WBC may indicate severe inflammation or infection, as well as increased blood viscosity, predisposing patients to secondary complications. High RBC and low Plt counts were also associated with an increased risk of complications.

A large-scale study on Chinese individuals found a negative correlation between plasma HDL and the risk of ischemic stroke, a weak positive correlation between TG levels and stroke risk, and a strong correlation between LDL and apolipoprotein B levels (*Sun et al., 2019*). High HDL levels are linked to better prognosis (*Bandeali and Farmer, 2012*). The data analysis and model interpretation results align with these findings, showing that high LDL, low HDL, and elevated TG levels are more likely to result in PSE. This can be explained by high cholesterol and TG levels increasing blood viscosity and contributing to vascular sclerosis, promoting clot formation (*Pitkänen et al., 2016*; *Gasparini et al.,*

*2022*; *Abraira et al., 2020*). Higher D-dimer levels indicate more significant brain tissue damage, increasing the likelihood of PSE. In general, lower APTT and fibrinogen levels are associated with higher PSE risk, while INR, PT, and TT have smaller impacts. Among liver function indicators, AST had the greatest influence on PSE. High AST, low ALT, and low albumin levels also had some impact. *Ding et al., 2023* reported that liver enzyme subgroups defined by ALT and AST were linked to higher risks of adverse outcomes, consistent with this findings.

Studies have shown that renal function biomarkers such as urinary microalbumin, cystatin C, and creatinine are associated with higher stroke recurrence rates and poorer prognosis (*Ding et al., 2023*). In the light of statistical results, low urea levels and high uric acid levels had a negative impact (*Zhang et al., 2023*; *Wang et al., 2019*; *Wang et al., 2021*). Elevated uric acid at admission was positively associated with PSE, although patients with a prior diagnosis of hyperuricemia were less likely to develop epilepsy. Since uric acid acts as a strong antioxidant and has neuroprotective properties (*Ng et al., 2017*), patients with normal liver and kidney function and mild hyperuricemia may exhibit greater resilience in emergencies (*Amaro et al., 2011*). However, excessively high uric acid levels suggest metabolic disorders and poor liver and kidney function, which are linked to a poor prognosis.

When stroke patients are admitted, cardiac enzyme tests are often performed to rule out myocardial ischemia. However, studies have shown that elevated CK-MB in stroke patients may not be solely heart-related (*Ay et al., 2002*). Cardiac enzymes are important prognostic indicators (*Liu et al., 2014*; *Zeng et al., 2021*) and have been incorporated into stroke scores (*Hijazi et al., 2016*). Some studies have reported a higher incidence of abnormal serum cardiac enzyme levels in the acute phase of stroke. Although these abnormalities are not related to stroke type, they are associated with stroke severity, with patients exhibiting consciousness disorders having a significantly higher incidence of abnormal cardiac enzymes than those without such disorders (*Zheng Yuan-Hui and Jian, 2009*). According to the statistical and SHAP results, CK, CK-MB, and IMA in the cardiac enzyme profile demonstrated significant impact and high predictive value, though further research is needed to understand the specific mechanisms involved (*Ng et al., 2017*).

Despite incorporating extensive clinical, imaging, and laboratory data to build more accurate prediction models using machine learning algorithms, surpassing traditional statistical methods, there were several limitations in the modeling process.

Although the current study offers valuable insights, the dataset may not be fully representative, and the model's generalizability requires further evaluation. The data were collected from multiple tertiary hospitals and included over 20,000 cases, but earlier data were lost due to hospital system upgrades. Consequently, the dataset primarily reflects patients diagnosed within the past five years and is predominantly from the Chongqing region, which may limit the model's applicability to other geographic areas.

Additionally, the retrospective nature of the study led to the absence of some important predictive indicators. Several potentially valuable features, such as hemorheology, thromboelastography, and hormone levels, were missing and had to be excluded, potentially affecting the model's accuracy. Including these features could further enhance the model's predictive power.

To improve the model, it would be beneficial to incorporate additional data beyond baseline patient characteristics. The current analysis mainly used results from the initial examination upon admission, without fully leveraging information from subsequent exams. Future research could employ recurrent neural networks to extract features from the entire sequence of examinations more comprehensively.

To strengthen this study further, data standardization should be improved, and the number of cases and key indicators should continue to grow. Additionally, exploring more advanced scientific methods, such as deep learning, and utilizing all available data could enhance prediction accuracy.

## Materials and methods
### Research patients
This study retrospectively included all stroke patients admitted to Chongqing Emergency Center between June 2017 and June 2022 to develop the prediction model. Data from three external validation centers—Qianjiang Central Hospital, Bishan District People's Hospital, and Yubei District Traditional Chinese Medicine Hospital—were collected between July 2022 and July 2023 to validate and

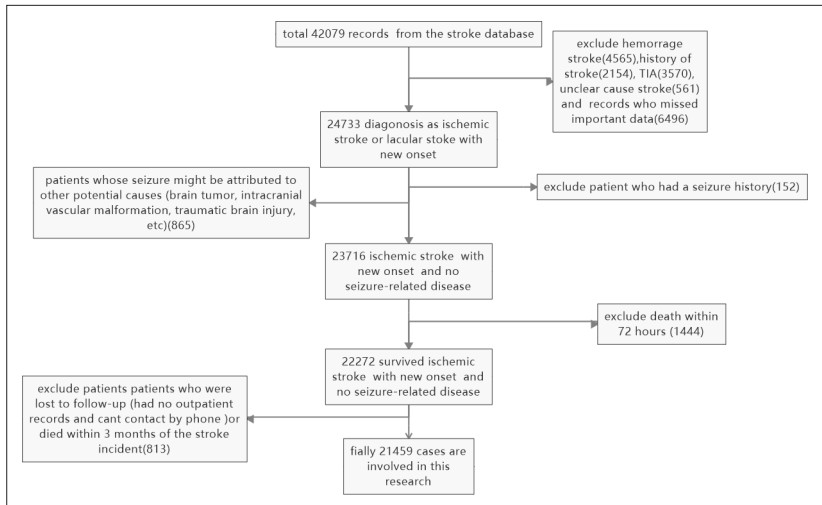

**Figure 4.** Slection and exclusion procedure of patients. A total of 42,079 records were retrieved from the stroke database, and 24,733 patients were diagnosed with ischemic or lacunar stroke with new onset. Hemorrhagic strokes (4565), a history of stroke (2154), TIA (3570), unclear cause strokes (561), and records with missing essential data (6496) were excluded. Patients whose seizures might have been caused by other factors (such as brain tumors, intracranial vascular malformations, or traumatic brain injury) (865), those with a seizure history (152), and patients who died in the hospital (1444) were also excluded. Additionally, patients lost to follow-up (those without outpatient records or unreachable by phone) or who died within three months of the stroke incident (813) were excluded. Finally, 21,459 cases were included in the study.

evaluate the model externally. The external validation cohort focused on collecting positive cases to accurately test the model's ability to identify them.

Inclusion criteria included: (1) Patients aged 18–90 years at admission; (2) Diagnosed with acute ischemic stroke and hospitalized for treatment.

Exclusion criteria were: (1) History of stroke or transient ischemic attack (TIA); (2) History of conditions such as traumatic brain injury, intracranial tumors, or cerebral vascular malformations that may cause epilepsy; (3) History of epilepsy or prior antiseizure medication use for seizure prevention or for other diseases (e.g. migraine or psychiatric disorders); (4) Death within 72 hr after stroke onset.

De-identified data from relevant patients were collected to build a multi-modal stroke patient database. The study protocol was approved by the Ethics Committees of Chongqing University Center Hospital, Chongqing University Qianjiang Central Hospital, Bishan District People's Hospital, and Yubei District Traditional Chinese Medicine Hospital.

The selection process is outlined in *Figure 4*. A total of 42,079 records were retrieved from the stroke database, and 24,733 patients were diagnosed with new-onset ischemic or lacunar stroke. Patients with hemorrhagic strokes (4565), a history of stroke (2154), TIA (3570), unclear-cause strokes (561), and those with missing essential data (6496) were excluded. Additionally, patients whose seizures may have been caused by other factors (such as brain tumors, intracranial vascular malformations, or traumatic brain injury) (865), those with a history of seizures (152), and those who died in the hospital (1444) were excluded. Patients lost to follow-up (those without outpatient records or unreachable by phone) or who died within three months of the stroke incident (813) were also excluded. In total, 21,459 cases were included in the study.

## Data collection

Relevant records and data were extracted from hospital databases. PostgreSQL was used to manage the data, with Structured Query Language (SQL) queries organized as follows:

1. General Information: This included gender, age, and NIH Stroke Scale (NIHSS) score at admission.
2. Comorbidities and Complications: Conditions such as uremia, deep vein thrombosis (DVT), diabetes mellitus, hypertension, coronary atherosclerosis, atrial fibrillation, cerebral hernia, hydrocephalus, hypoproteinemia, hyperuricemia, hyperlipidemia, internal carotid stenosis, and common carotid stenosis were recorded.

3. Brain Involvement (CT or MRI records): Involvement of cortical lobes (frontal, parietal, temporal, occipital, and insular) and subcortical areas (basal ganglia, internal capsule, brain stem, cerebellum, periventricular area, centrum semiovale, and thalamus) was noted. Cortical involvement was scored with each lobe contributing 1 point, and subcortical involvement was scored similarly, with each area contributing 1 point.
4. Vascular Involvement (CTA, MRA, or DSA records): The presence of vascular stenosis or occlusion in the anterior cerebral artery (ACA), middle cerebral artery (MCA), posterior cerebral artery (PCA), vertebral artery (VA), and basilar artery (BA) was documented.
5. Key Laboratory Indicators: These included blood lipid levels such as triglycerides, high-density lipoprotein cholesterol (HDL), and low-density lipoprotein cholesterol (LDL); liver function markers such as alanine transaminase (ALT), aspartate aminotransferase (AST), bilirubin, and albumin; renal function markers such as urea, blood uric acid (BUA), and creatinine; blood gas parameters such as lactate, anion gap, and total carbon dioxide (TCO2); coagulation markers such as international normalized ratio (INR), prothrombin time (PT), activated partial thromboplastin time (APTT), thrombin time (TT), D-dimer, and fibrinogen; and myocardial enzymes such as creatine kinase (CK), creatine kinase isoenzyme (CK-MB), lactate dehydrogenase (LDH), ischemic modified albumin (IMA), and α-hydroxybutyrate dehydrogenase (HBDH).

## Data processing and model building

Processing of Missing Data: Laboratory indicators were recorded from the first set of tests after stroke admission. Indicators with more than 10% missing data were excluded. Remaining indicators with missing values were imputed using the random forest algorithm with default parameters. Features were processed in order of increasing missing data to minimize imputation complexity. During imputation, missing values in other features were temporarily replaced with 0, and predicted values were inserted into the original feature matrix before moving to the next feature. This process continued until all features were complete.

Distribution of Characteristics: Univariate analysis was performed to compare the distribution of characteristics between PSE-negative and PSE-positive groups. The dataset was then divided into a training set and a test set in a 7:3 ratio.

Handling Imbalanced Data: Due to the low incidence of PSE and the small proportion of positive cases, positive data in the training set were augmented using the Synthetic Minority Over-sampling Technique combined with Edited Nearest Neighbors (SMOTEENN). The SMOTEENN method from the imblearn Python package was applied with default parameters, and a random seed of 42 was set for reproducibility.

Processing of Categorical Data: Categorical variables were transformed using one-hot encoding. The LASSO method was then applied to the training set to identify the most important features.

Model Building: LASSO regression was used to select the 20 most important features. Nine widely used machine learning methods were employed, including Naive Bayes, Logistic Regression, Decision Tree, Random Forest, Gradient Boosting, Multi-Layer Perceptron, XGBoost, LightGBM, and K-Nearest Neighbors. Hyperparameters were optimized through grid search to enhance model performance. Evaluation metrics included accuracy, sensitivity, specificity, F1-score, positive predictive value, and negative predictive value. Additionally, ROC curves, calibration curves, and decision curves were generated to assess model performance. An independent external validation dataset was used to evaluate the model's generalizability. The SHAP algorithm was then applied to the best-performing model to interpret feature contributions and their clinical relevance. This approach enabled the development of a robust machine learning model with strong predictive performance and interpretability, providing valuable support for clinical decision-making.

## Statistical approach

PostgreSQL v15 (http://www.postgresql.org/) was used to search and extract data from the local database. Statistical analysis was performed using the open-source Scipy.stats package in Python. The details of the univariate significance analysis were as follows:

The Shapiro-Wilk test was used to assess the normality of each feature's distribution. For features not following a normal distribution, the Mann-Whitney U test was used to evaluate their significance in relation to the target variable. For features with a normal distribution, the Levene test was performed

to evaluate the homogeneity of variances. Features with homogeneous variances were analyzed using the Student's t-test, while those with heterogeneous variances were analyzed using Welch's t-test.

Confidence intervals for AUC values and Brier scores were calculated using 1000 bootstrap resampling iterations. Binary classification thresholds for predicted probabilities were set using the maximum Youden index derived from the training cohort. A two-tailed p-value of less than 0.05 was considered statistically significant throughout the study.

All code used in this study is available at https://github.com/conanan/lasso-ml (copy archived at *conanan, 2024*).

## Conclusion

An interpretable machine learning model was developed to predict the risk of PSE in hospitalized patients with ischemic stroke. Using a large dataset of medical records, the model demonstrated strong predictive performance for PSE. Key predictors identified by the model include NIHSS, D-dimer, lactate, and WBC, along with liver function and cardiac enzyme profile indicators. The model's transparency and interpretability can build trust among clinicians and support decision-making. While the results are promising, further prospective studies are necessary to validate the clinical utility of this tool before it can be applied in real-world settings.

## Acknowledgements

The authors would like to thank their colleagues in the information and imaging departments for their hard work contributing to the final research results.

## Additional information

### Funding

| Funder | Grant reference number | Author |
| --- | --- | --- |
| Central University Basic Research Fund of China | 2023CDJYGRH-ZD06 | Yongbing Deng |
| Emergency Medicine chongqing Key Laboratory Talent Innovation and development joint fund project | 2024RCCX10 | Yongbing Deng |

The funders had no role in study design, data collection and interpretation, or the decision to submit the work for publication.

### Author contributions

Jinxin Liu, Data curation, Software, Formal analysis, Investigation, Visualization, Methodology, Writing – original draft, Writing – review and editing; Haoyue He, Data curation, Software, Formal analysis, Visualization, Methodology, Writing – original draft, Writing – review and editing; Yanglingxi Wang, Writing – original draft, Writing – review and editing; Jun Du, Kaixin Liang, Data curation, Methodology, Writing – original draft, Writing – review and editing; Jun Xue, Data curation, Writing – original draft, Writing – review and editing; Yidan Liang, Peng Chen, Formal analysis, Methodology, Writing – original draft, Writing – review and editing; Shanshan Tian, Data curation, Formal analysis, Supervision, Methodology, Writing – original draft, Writing – review and editing; Yongbing Deng, Conceptualization, Resources, Data curation, Formal analysis, Supervision, Funding acquisition, Validation, Visualization, Methodology, Writing – original draft, Writing – review and editing

### Author ORCIDs

Jinxin Liu (iD) https://orcid.org/0009-0003-8923-2536
Yongbing Deng (iD) http://orcid.org/0000-0002-8581-5748

Reviewer #1 (Public review): https://doi.org/10.7554/eLife.98759.3.sa1

Reviewer #2 (Public review): https://doi.org/10.7554/eLife.98759.3.sa2
Reviewer #3 (Public review): https://doi.org/10.7554/eLife.98759.3.sa3
Author response https://doi.org/10.7554/eLife.98759.3.sa4

## Additional files

### Supplementary files

• MDAR checklist

• Supplementary file 1. Single factor significant analysis results. This table presents the results of the Chi-Square and Mann-Whitney U tests used to evaluate the association of various features with positive and negative samples. Sample sizes: Positive samples (n=954) and negative samples (n=20,789). Statistical methods: The Chi-Square test assesses the relationship between categorical variables, while the Mann-Whitney U test compares differences between independent groups for continuous data. p-values: Indicate the significance of the associations, with lower values suggesting stronger evidence against the null hypothesis. Statistical values: Include counts and percentages of samples for each feature in both groups, along with the calculated statistics for each test.

• Supplementary file 2. Single factor significant analysis results. This table presents the results of a single-factor significance analysis for various features across two groups of samples: negative samples (0) and positive samples (1). Sample size: Group 0 (Negative): n=20,789 Group 1 (Positive): n=954 Feature analysis: For each feature, the table includes the mean and standard deviation (±) for both groups, odds ratios (OR) from univariable analysis, coefficients (coef), standard errors (std err), z-scores (z), p-values (p>|z|), and 95% confidence intervals ([0.025, 0.975]). Significance levels: Features with statistically significant differences are indicated by p-values less than 0.05. An odds ratio greater than 1 suggests an increased risk associated with the feature in the positive group, while an odds ratio less than 1 suggests a decreased risk. Labels: The last two columns provide the proportions of the positive and negative samples for selected features.

• Supplementary file 3. Multivariable analysis results. This table presents the results of a multivariable analysis for various features across two groups of samples: negative samples (0) and positive samples (1). Sample size: Group 0 (Negative): n=20,789 Group 1 (Positive): n = 954 Feature analysis: For each feature, the table includes the mean and standard deviation (±) for both groups, odds ratios (OR) from multivariable analysis, coefficients (Coef.), standard errors (Std. Err.), z-scores (z), p-values (p>|z|), and 95% confidence intervals ([0.025, 0.975]). Significance levels: Features with statistically significant differences are indicated by p-values less than 0.05. An odds ratio greater than 1 indicates an increased risk associated with the feature in the positive group, while an odds ratio less than 1 suggests a decreased risk. Labels: The last column presents the proportions of the positive and negative samples for selected features.

• Supplementary file 4. Model performance evaluation results. This table presents the performance evaluation metrics for various machine learning models, including AUC, Accuracy, Sensitivity (Recall), Specificity, F1-score, Positive Predictive Value (PPV/Precision), and Negative Predictive Value (NPV). AUC: Area Under the Curve, indicating the model's ability to distinguish between positive and negative samples; values closer to 1 indicate better performance. Accuracy: The proportion of correctly classified samples among the total samples. Sensitivity/Recall: The proportion of correctly identified positive samples out of all actual positive samples. Specificity: The proportion of correctly identified negative samples out of all actual negative samples. F1-score: The harmonic mean of precision and recall, considering both the accuracy and completeness of the model. PPV/Precision: The proportion of correctly identified positive samples among all samples predicted as positive. NPV: The proportion of correctly identified negative samples among all samples predicted as negative.

• Source code 1. Statistical analysis notebook. This Jupyter notebook conducts statistical analyses of the results obtained from the machine learning models, providing insights and summaries. It includes the original data for Tables 1, 2, and 3, which are related to statistical and regression analysis.

• Source code 2. Data preprocessing script. This Python script is dedicated to data preprocessing tasks, specifically designed to fill missing data using Random Forest (RF) imputation techniques, ensuring data integrity for subsequent analyses.

• Source code 3. Lasso regression, model construction, and SHAP analysis. This main notebook implements Lasso regression, along with model construction and SHAP (SHapley Additive

exPlanations) analysis for interpretability. It contains the original Figures 2, 3, 4, and Table 4, which relate to model performance and interpretation.

• Source code 4. Cross-validation for Lasso regression. This notebook is designed to perform fivefold cross-validation for the Lasso regression and model construction, ensuring a robust evaluation of model performance. It includes figures and tables that summarize the results of the cross-validation process.

• Source code 5. External model testing. This notebook contains code for external testing of the model, evaluating its performance on new external data. All relevant data related to the evaluation process is included within this code.

### Data availability

The codes, models, analysis and results are uploaded at https://github.com/conanan/lasso-ml (copy archived at *conanan, 2024*). The full dataset and codes are also uploaded at https://doi.org/10.5061/dryad.w0vt4b92c.

The following dataset was generated:

| Author(s) | Year | Dataset title | Dataset URL | Database and Identifier |
|---|---|---|---|---|
| Deng Y | 2024 | Data From: Predictive Models for Secondary Epilepsy in Patients with Acute Ischemic Stroke Within One Year | https://doi.org/10.5061/dryad.w0vt4b92c | Dryad Digital Repository, 10.5061/dryad.w0vt4b92c |

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
