## [Editor Report · eLife Assessment]

This **valuable** study reports machine learning models derived from large-scale data to predict the risk of post-stroke epilepsy. The evidence supporting the conclusions is **convincing**, although there are some validation issues (lack of cross-validation, possible bias in external validation results). The study may be of interest in the field of clinical neurology

---

## [Referee Report · Reviewer #1 (Public review)]

Summary:

This is a large cohort of ischemic stroke patients from a single centre. The author successfully set up predictive models for PTS.

Strengths:

The design and implementation of the trial are acceptable, with the credibility of the results. It may provide evidence of seizure prevention in the field of stroke treatment.

Weaknesses:

My concerns are well responded to.

---

## [Referee Report · Reviewer #2 (Public review)]

Summary

The authors present multiple machine-learning methodologies to predict post-stroke epilepsy (PSE) from admission clinical data.

Strengths

The Statistical Approach section is very well written. The approaches used in this section are very sensible for the data in question.

Typos have now been addressed and improved interpretability has been added to the paper, which is appreciated.

Weaknesses

The authors have clarified that the first features available for each patient have been used. However, they have not shown that these features did not occur before the time of post-stroke epilepsy. Explicit clarification of this should be performed.

The likely impact of the work on the field

If this model works as claimed, it will be useful for predicting PSE. This has some direct clinical utility.

Analysis of features contributing to PSE may provide clinical researchers with ideas for further research on the underlying aetiology of PSE.

---

## [Referee Report · Reviewer #3 (Public review)]

Summary:

The authors report the performance of a series of machine learning models inferred from a large-scale dataset and externally validated with an independent cohort of patients, to predict the risk of post-stroke epilepsy. Some of the reported models have very good explicative performance, and seem to have very good predictive ability.

Strengths:

The models have been derived from real-world large-scale data.

Performances of the best-performing models seem to be very good according to the external validation results.

Early prediction of risk of post-stroke epilepsy would be of high interest to implement early therapeutic interventions that could improve prognosis.

Code is publicly available. The authors also stated that the datasets used are available on request.

Weaknesses:

The writing of the article may be significantly improved.

Although the external validation is appreciated, cross-validation to check robustness of the models would also be welcome.

External validation results may be biased/overoptimistic, since the authors informed that "The external validation cohort focused more on collecting positive cases 80 to examine the model's ability to identify positive samples", which may result in overoptimistic PPV and Sensitivity estimations. The specificity for the external validation set has not been disclosed.

---

## [Author Response]

The following is the authors’ response to the current reviews.

**Public Reviews:**

**Reviewer #2 (Public review):**
Weaknesses:The authors have clarified that the first features available for each patient have been used. However, they have not shown that these features did not occur before the time of post-stroke epilepsy. Explicit clarification of this should be performed.

The data utilized in our analysis were collected during the first examination or test conducted after the patients' admission. We specifically excluded any patients with a history of epilepsy, ensuring that all cases of epilepsy identified in our study occurred after admission. Therefore, the features we analyzed were collected after the patients' admission but prior to the onset of post-stroke epilepsy.

**Reviewer #3 (Public review):**
Weaknesses:The writing of the article may be significantly improved.Although the external validation is appreciated, cross-validation to check robustness of the models would also be welcome.

Thank you for your helpful advice. Performing n-fold cross-validation is a crucial step to ensure the reliability and robustness of the reported results, especially when dealing with the datasets which don't have sufficient quantity. We revised our code and did a 5 fold cross-validation version ,it didn’t have much promote（because our model has reach the auc of 0.99）.Considering that we have sufficient quantity of more than 20000 records, we think split the dataset by 7:3 and train the model is enough for us. We have uploaded the code of 5 fold cross-validation version and ploted the 5 fold test roc on GitHub at https://github.com/conanan/lasso-ml/lasso_ml_cross_validation.ipynb as an external resource. We trained the 5 fold average model and ploted the 5 fold test roc curves, the results show some improvement, but it is not substantial because the best model are still tree models in the end.

External validation results may be biased/overoptimistic, since the authors informed that "The external validation cohort focused more on collecting positive cases 80 to examine the model's ability to identify positive samples", which may result in overoptimistic PPV and Sensitivity estimations. The specificity for the external validation set has not been disclosed.

Thank you for your valuable feedback regarding the external validation results. We appreciate your concerns about potential bias and overoptimism in our estimations of positive predictive value (PPV) and sensitivity.

To clarify, we have uploaded the code for external validation on GitHub at https://github.com/conanan/lasso-ml. The results indicate that the PPV is 0.95 and the specificity is 0.98.

While we focused on collecting more positive cases due to their lower occurrence rate, this approach allows us to better evaluate the model's ability to predict positive samples, which is crucial in clinical settings. We believe that emphasizing positive cases enhances the model's utility for practical applications (So a little overoptimism is acceptable).

The following is the authors’ response to the original reviews.

**Public Reviews:**

**Reviewer #1 (Public Review):**
Weaknesses 1:The methodology needs further consideration. The Discussion needs extensive rewriting.

Thanks for your advice, we have revised the Discussion

**Reviewer #2 (Public Review):**
Weaknesses 2:There are many typos and unclear statements throughout the paper.There are some issues with SHAP interpretation. SHAP in its default form, does not provide robust statistical guarantees of effect size. There is a claim that "SHAP analysis showed that white blood cell count had the greatest impact among the routine blood test parameters". This is a difficult claim to make.

Thank you for your suggestion that the SHAP analysis is really just a means of interpreting the model. In our research, we compared the SHAP analysis with traditional statistical methods, such as regression analysis. We found the SHAP results to be consistent with the statistical results from the regression for variables like white blood cell count (see Table 1). This alignment leads us to believe the SHAP analysis is providing reliable insights in this context

The Data Collection section is very poorly written, and the methodology is not clear.

Thanks for your advice, we have revised the Data Collection section.

There is no information about hyperparameter selection for models or whether a hyperparameter search was performed. Given this, it is difficult to conclude whether one machine learning model performs better than others on this task.

Thank you for the advices of performing hyperparameter. We used the package of sklearn, xgboost, lightgbm of python 3.10 to construct the model and didn’t change the default settings before. It is not proper and may lead to less certain conclusions. Now we carry out grid search to select and optimize hyperparameters and they make the model better. The best model is still RF.

The inclusion and exclusion criteria are unclear - how many patients were excluded and for what reasons?

The procedure of selection is in figure1. Total there are 42079 records from the stroke database, 24733 patients were diagnosed as ischemic stroke or lacular stoke with new onset. Then we excluded hemorrage stroke (4565),history of stroke (2154), TIA (3570), unclear cause stroke (561) and records who missed important data (6496). Then we excluded patients whose seizure might be attributed to other potential causes (brain tumor, intracranial vascular malformation, traumatic brain injury,etc) (865). Then we exclude patient who had a seizure history (152) or died in hospital (1444). Then we excluded patients who were lost in follow-up (had no outpatient records and can’t contact by phone) or died within 3 months of the stroke incident (813). Finally 21459 cases are involved in this research.

There is no sensitivity analysis of the SMOTE methodology: How many synthetic data points were created, and how does the number of synthetic data points affect classification accuracy?

Thanks for your remind, we have accept these advice and change the SMOTE to SMOTEENN (Synthetic Minority Over-sampling Technique combined with Edited Nearest Neighbors) technique to resample an imbalanced dataset for machine learning. The code is

smoteenn = SMOTEENN(samplingstrategy='auto', randomstate=42)

the SMOTEENN class comes from the imblearn library. The samplingstrategy='auto' parameter tells the algorithm to automatically determine the appropriate sampling strategy based on the class distribution. The randomstate=42 parameter sets a seed for the random number generator, ensuring reproducibility of the results.

Did the authors achieve their aims? Do the results support their conclusions?

Yes, we have achieve some of the aims of predicting PSE while still leave some problem.

The paper does not clarify the features' temporal origins. If some features were not recorded on admission to the hospital but were recorded after PSE occurred, there would be temporal leakage.

The data used in our analysis is from the first examination or test conducted after the patients' admission, retrieved from a PostgreSQL database. First, we extracted the initial admission date for patients admitted due to stroke. Then, we identified the nearest subsequent examination data for each of those patients.

The sql code like follows:

SELECT TO_DATE(condition_start_date, 'DD-MM-YYYY') AS DATE

FROM diagnosis

WHERE person_id = {} and (condition_name like '%梗死%' or condition_name like '%梗塞%') and (condition_name like '%脑%'or condition_name like '%腔隙%')

order by DATE limit 1

The authors claim that their models can predict PSE. To believe this claim, seeing more information on out-of-distribution generalisation performance would be helpful. There is limited reporting on the external validation cohort relative to the reporting on train and test data.

Thank you for the advice. The external validation is certainly very important, but there have been some difficulties in reaching a perfect solution. We have tried using open-source databases like the MIMIC database, but the data there does not fit our needs as closely as the records from our own hospital. The MIMIC database lacks some of the key features we require, and also lacks the detailed patient follow-up information that is crucial for our analysis. Given these limitations, we have decided to collect newer records from the same hospitals here in Chongqing. We believe this will allow us to build a more comprehensive dataset to support robust external validation. While it may not be a perfect solution, gathering this additional data from our local healthcare system is a pragmatic step forward. Looking ahead, we plan to continue expanding this Chongqing-based dataset and report on the results of the greater external validation in the future. We are committed to overcoming the challenges around data availability to strengthen the validity and generalizability of our research findings.

For greater certainty on all reported results, it would be most appropriate to perform n-fold cross-validation, and report mean scores and confidence intervals across the cross-validation splits

Thank you for your helpful advice. Performing n-fold cross-validation is a crucial step to ensure the reliability and robustness of the reported results, especially when dealing with the datasets which don't have sufficient quantity. While we have sufficient quantity of more than 20000 records, so we think split the dataset by 7:3 and train the model is enough for us. We revised our code and did a 5 fold cross-validation version ,it had little promote（because our model has reach the auc of 0.99）, we may use this great technique in our next study if there is not enough cases.

Additional context that might help readersThe authors show force plots and decision plots from SHAP values. These plots are non-trivial to interpret, and the authors should include an explanation of how to interpret them.

Thank you for your helpful advice. It is a great improve for our draft, we have added the explanation that we use the force plot of the first person to show the influence of different features of the first person, we can see that long APTT time contribute best to PSE, then the AST level and others, the NIHSS score may be low and contribute opposite to the final result. Then the decision plot is a collection of model decisions that show how complex models arrive at their predictions

**Reviewer #3 (Public Review):**
Weaknesses3:There are issues with the readability of the paper. Many abbreviations are not introduced properly and sometimes are written inconsistently. A lot of relevant references are omitted. The methodological descriptions are extremely brief and, sometimes, incomplete.

Thanks for your advice, we have revised these flaws.

The dataset is not disclosed, and neither is the code (although the code is made available upon request). For the sake of reproducibility, unless any bioethical concerns impede it, it would be good to have these data disclosed.

Thank you for your recommendations. We have made the code available on GitHub at https://github.com/conanan/lasso-ml. While the data is private and belongs to the hospital. Access can be requested by contacting the corresponding author to apply from the hospitals and specifying the purpose of inquiry.

Although the external validation is appreciated, cross-validation to check the robustness of the models would also be welcome.

Thank you for your valuable advice. Performing n-fold cross-validation is crucial for ensuring the reliability and robustness of results, especially with limited datasets. However, since we have over 20,000 records, we believe that a 70:30 split for training and testing is sufficient.

We revised our code and implemented 5-fold cross-validation, which provided minimal improvement, as our model has already achieved an AUC of 0.99. We plan to use this technique in future studies if we encounter fewer cases.

**Recommendations for the authors:**

**Reviewer #1 (Recommendations For The Authors):**
My comments include two parts:(1) Methodologya-This study was based on multiple clinical indicators to construct a model for predicting the occurrence of PSE. It involved various multi-class indicators such as the affected cortical regions, locations of vascular occlusion, NIHSS scores, etc. Only using the SHAP index to explain the impact of multi-class variables on the dependent variable seems slightly insufficient. It might be worth considering the use of dummy variables to improve the model's accuracy.

Thank you for the detailed feedback on the study methodology. The SHAP analysis is really just a means of interpreting the model, which we compared with the combination of SHAP and traditional statistics, so we think SHAP analysis is reliable in this research. We have used the dummy variables, expecially when dealing with the affected cortical regions, locations of vascular occlusion, for example if frontal region is involved the variable is 1. But they have less impact in the machine learning model

b-The study used Lasso regression to select 20 features to build the model. How was the optimal number of 20 features determined?

Lasso regression is a commonly used feature screening method. Since we extract information from the database and try to include as many features as possible, the cross-verification curve of lasso regression includes 78 features best, but it will lead to too complex model. We select 10,15,20,25,30 features for modeling according to the experiment. When 20 features are found, the model parameters are good and relatively concise. Improve the number of features contribute little to the model effect, decrease the number of features influence the concise of model ,for example the auc of the model with 15 features will drop under 0.95. So we finally select 20 features.

c-The study indicated that the incidence rate of PSE in the enrolled patients is 4.3%, showing a highly imbalanced dataset. If singly using the SMOTE method for oversampling, could this lead to overfitting?

Thanks for your remind, singly using the SMOTE method for oversampling is inproper. Now we have find this improvement and change the SMOTE to SMOTEENN (Synthetic Minority Over-sampling Technique combined with Edited Nearest Neighbors) technique to resample an imbalanced dataset for machine learning. First, oversampling with SMOTE and then undersampling with ENN to remove possible noise and duplicate samples. The code is

smoteenn = SMOTEENN(sampling_strategy='auto', random_state=42)

the SMOTEENN class comes from the imblearn library. The sampling_strategy='auto' parameter tells the algorithm to automatically determine the appropriate sampling strategy based on the class distribution. The random_state=42 parameter sets a seed for the random number generator, ensuring reproducibility of the results.

(2) Clinical aspects:Line 8, history of ischemic stroke, this is misexpression, could be: diagnosis of ischemic stroke.Line 8, several hospitals, should be more exact; how many?Line 74 indicates that the data are from a single centre, this should be clarified.Line 4 data collection: The criteria read unclear; please clarify further.

Thanks for your remind, we have revised the draft and correct these errors.

Line 110, lab parameters: Why is there no blood glucose?

Because many patients' blood sugar fluctuates greatly and is easily affected by drugs or diet, we finally consider HBA1c as a reference index by asking experts which is more stable.

Line 295, The author indicated that data lost; this should be clarified in the results part, and further, the treatment of missing data should be clarified in the method part.

Thanks for your remind, we have revised the draft and correct these errors.

I hope to see a table of the cohort's baseline characters. The discussion needs extensive rewriting; the author seems to be swinging from the stoke outcome and the seizure, sometimes losing the target.

Figure1 is the procedure of the selection of patients. Table1 contains the cohort's baseline characters

For the swinging from the stoke outcome and the seizure, that is because there are few articles on predicting epilepsy directly by relevant indicators, while there are more articles on prognosis. So we can only take epilepsy as an important factor in prognosis and comprehensively discuss it, or we can't find enough articles and discuss them

**Reviewer #2 (Recommendations For The Authors):**
There are typos and examples of text that are not clear, including:"About the nihss score, the higher the nihss score, the more likely to be PSE, nihss score has a third effect just below white blood cell count and D-dimer.""and only 8 people made incorrect predictions, demonstratijmng a good predictive ability of the model.""female were prone to PSE"" Waafi's research""One-heat' (should be one-hot)

Thanks for your remind, we have revised the draft and correct these errors.

The Data Collection section is poorly written, and the methodology is not clear. It would be much more appropriate to include a table of all features used and an explanation of what these features involve. It would also be useful to see the mean values of these features to assess whether the feature values are reasonable for the dataset.

Thanks for your remind. All data are from the first examination or test after admission, presented through the postgresql database . First we extract the first date of the patients who was admitted by stroke ,then we extract informations from the nearest examination from the admission. We extract by the SQL code by computer instead of others who may extract data by manual so we get as much data as possible other than only get the features which was reported before .The table of all features used and their mean±std is in table1.

The paper does not clarify the features' temporal origins. If some features were not recorded on admission to the hospital but were recorded after PSE occurred, there would be temporal leakage. I would need this clarified before believing the authors achieved their claims of building a predictive model.

All relevant index results were from the first examination after admission, and the mean standard deviation was listed in the statistical analysis section in table1.

The authors claim that their models can predict PSE. To believe this claim, seeing more information on out-of-distribution generalisation performance would be helpful. There is limited reporting on the external validation cohort relative to the reporting on train and test data.

Thank you for the advice, the external validation is very important but there are some difficulties to reach a perfect one. We have tried some of the open source database like the mimic database ,but these data don't fit our request because they don't have as much features as our hospital and lack of follow-up of the relevant patients. In the end we collected the newer records in the same hospitals in Chongqing and we will collect more and report a greater external validation in the future.

For greater certainty on all reported results, It would be most appropriate to perform n-fold cross-validation, and report mean scores and confidence intervals across the cross-validation splits.

Thank you for your helpful advice. Performing n-fold cross-validation is a crucial step to ensure the reliability and robustness of the reported results, especially when dealing with the datasets which don't have sufficient quantity. While we have sufficient quantity of more than 20000 records, so we think split the dataset by 7:3 and train the model is enough for us. We revised our code and did a 5 fold cross-validation version ,it had little promote, we will use this great technique in our next study.

The authors show force plots and decision plots from SHAP values. These plots are non-trivial to interpret, and the authors should include an explanation of how to interpret them.

It is a great improve for our draft, we have added the explanation we use the force plot of the first person to show the influence of different features of the first person, we can see that long APTT time contribute best to PSE, then the AST level and others, the NIHSS score may be low and contribute lower to the final result. Then the decision plot is a collection of model decisions that show how complex models arrive at their predictions

**Reviewer #3 (Recommendations For The Authors):**
Abbreviations should not be defined in the abstract or only in the abstract.Please explicit what are the purposes of the study you are referring to in "Currently, most studies utilize clinical data to establish statistical models, survival analysis and cox regression."Authors affirm: "there is still a relative scarcity of research 49 on PSE prediction, with most studies focusing on the analysis of specific or certain risk factors ." This statement is especially curious since the current study uses risk factors as predictors.It is not clear to me what the authors mean by "No study has proposed or established a more comprehensive and scientifically accurate prediction model." The authors do not summarize the statistical parameters of previously reported model, or other relevant data to assess coverage or validity maybe including a Table summarizing such information would be appropriate. In any case, I would try to omit statements that imply, to some extent, discrediting previous studies without sufficient foundation."antiepileptic drugs" is an outdated name. Please use "antiseizure medications"

Thanks for your remind, we have revised the draft and correct these errors.

The authors say regarding missing data that they "filled the data of the remaining indicators with missing values of more than 1000 cases by random forest algorithm". Please clarify what you mean by "of more than 1000 cases." Also, provide details on the RF model used to fill in missing data.

Thanks for your remind. "of more than 1000 cases" was a wrong sentence and we have corrected it. Here is the procedure, first we counted the values of all laboratory indicators for the first time after stroke admission (everyone who was admitted because of stroke would perform blood routine , liver and kidney function and so on), excluded indicators with missing values of more than 10%, and filled the data of the remaining indicators with missing values by random forest algorithm using the default parameter. First, we go through all the features, starting with the one with the least missing (since the least accurate information is needed to fill in the feature with the least missing). When filling in a feature, replace the missing value of the other feature with 0. Each time a regression prediction is completed, the predicted value is placed in the original feature matrix and the next feature is filled in. After going through all the features, the data filling is complete.

Please specify what do you mean by negative group and positive group, Avoid tacit assumptions.

Thanks for your remind, we have revised the draft and correct these errors.

Please provide more details (and references) on the smote oversampling method. Indicate any relevant parameters/hyperparameters.

Thanks for your remind, we have accept these advice and change the SMOTE to SMOTEENN (Synthetic Minority Over-sampling Technique combined with Edited Nearest Neighbors) technique to resample an imbalanced dataset for machine learning. The code is

smoteenn = SMOTEENN(sampling_strategy='auto', random_state=42)

the SMOTEENN class comes from the imblearn library. The sampling_strategy='auto' parameter tells the algorithm to automatically determine the appropriate sampling strategy based on the class distribution. The random_state=42 parameter sets a seed for the random number generator, ensuring reproducibility of the results.

The methodology is presented in an extremely succinct and non-organic manner (e.g., Model building) Select the 20 features with the largest absolute value of LASSO." Please try to improve the narrative.

Lasso regression is a commonly used feature screening method. Since we extract information from the database and try to include as many features as possible, the cross-verification curve of lasso regression includes 78 features best, but it will lead to too complex model. We select 10,15,20,25,30 features for modeling according to the experiment. When 20 features are found, the model parameters are good and relatively concise. Improve the number of features contribute little to the model effect, decrease the number of features influence the concise of model ,for example the auc of the model with 15 features will drop under 0.95. So we finally select 20 features.

Many passages of the text need references. For example, those that refer to Levene test, Welch's t-test, Brier score, Youden index, and many others (e.g., NIHSS score). Please revise carefully.

Thanks for your remind, we have revised the draft and correct these errors.

"Statistical details of the clinical characteristics of the patients are provided in the table." Which table? Number?

Thanks for your remind, we have revised the draft and correct these errors， it is in table1.

Many abbreviations are not properly presented and defined in the text, e.g., wbc count, hba1c, crp, tg, ast, alt, bilirubin, bua, aptt, tt, d_dimer, ck. Whereas I can guess the meaning, do not assume everyone will. Avoid assumptions.ROC is sometimes written "ROC" and others, "roc." The same happens for PPV/ppv, and many other words (SMOTE; NIHSS score, etc.).Please rephrase "ppv value of random forest is the highest, reaching 0.977, which is more accurate for the identification of positive patients(the most important function of our models).". PPV always refer to positive predictions that are corroborated, so the sentences seem redundant.

Thanks for your remind, we have revised the draft and correct these errors.

What do you mean by "Complex algorithms". Please try to be as explicit as possible. The text looks rather cryptic or vague in many passages.

Thanks for your remind, "Complex algorithms" is corrected by machine learning.

The text needs a thorough English language-focused revision, since the sense of some sentences is really misleading. For instance "only 8 people made incorrect predictions,". I guess the authors try to say that the best algorithm only mispredicted 8 cases since no people are making predictions here. Also, regarding that quote... Are the authors still speaking of the results of the random forest model, which was said to be one of the best performances?

Thanks for your remind, we have revised the draft and correct these errors.

The authors say that they used, as predictors "comprehensive clinical data, imaging data, laboratory test data, and other data from stroke patients". However, the total pool of predictors is not clear to me at this point. Please make it explicit and avoid abbreviations.

Thanks for your remind, we have revised the draft and correct these errors.

Although the authors say that their code is available upon request, I think it would be better to have it published in an appropriate repository.

Thanks for your remind, we showed our code at https://github.com/conanan/lasso-ml.